

# A BERT-Span model for Chinese named entity recognition in rehabilitation medicine

Jinhong Zhong*, Zhanxiang Xuan*, Kang Wang and Zhou Cheng

School of Management, Hefei University of Technology, Hefei, Anhui, China
* These authors contributed equally to this work.

## ABSTRACT

**Background:** Due to various factors such as the increasing aging of the population and the upgrading of people's health consumption needs, the demand group for rehabilitation medical care is expanding. Currently, China's rehabilitation medical care encounters several challenges, such as inadequate awareness and a scarcity of skilled professionals. Enhancing public awareness about rehabilitation and improving the quality of rehabilitation services are particularly crucial. Named entity recognition is an essential first step in information processing as it enables the automated extraction of rehabilitation medical entities. These entities play a crucial role in subsequent tasks, including information decision systems and the construction of medical knowledge graphs.

**Methods:** In order to accomplish this objective, we construct the BERT-Span model to complete the Chinese rehabilitation medicine named entity recognition task. First, we collect rehabilitation information from multiple sources to build a *corpus* in the field of rehabilitation medicine, and fine-tune Bidirectional Encoder Representation from Transformers (BERT) with the rehabilitation medicine *corpus*. For the rehabilitation medicine *corpus*, we use BERT to extract the feature vectors of rehabilitation medicine entities in the text, and use the span model to complete the annotation of rehabilitation medicine entities.

**Result:** Compared to existing baseline models, our model achieved the highest F1 value for the named entity recognition task in the rehabilitation medicine *corpus*. The experimental results demonstrate that our method outperforms in recognizing both long medical entities and nested medical entities in rehabilitation medical texts.

**Conclusion:** The BERT-Span model can effectively identify and extract entity knowledge in the field of rehabilitation medicine in China, which supports the construction of the knowledge graph of rehabilitation medicine and the development of the decision-making system of rehabilitation medicine.

Corresponding author
Zhanxiang Xuan,
xuanzhanxiang@163.com

## INTRODUCTION

With the steady growth of the economy and the continuous improvement of living standards, there is an escalating demand to enhance the quality of life for individuals. In addition to disease prevention, clinical treatment, and physical health care, rehabilitation medicine is receiving growing attention as a crucial approach to assist patients with

disabilities in regaining their daily living abilities and enhancing their overall quality of life. According to statistics from a study by The Lancet (*Feng et al., 2020*), China has the highest rehabilitation needs globally, totaling 460 million people. According to the China Health and Wellness Statistical Yearbook from 2012 to 2020 (*Committee NHaW, 2012–2020*), the number of outpatient visits to rehabilitation hospitals in China surged from more than six million to more than 11 million. The increasing demand for rehabilitation medical care in China is evident. However, China's rehabilitation medical facilities suffer from inadequate resources, with many hospitals lacking dedicated rehabilitation medicine departments. Additionally, there is a shortage of professional rehabilitation physicians, with an average of less than six rehabilitation physicians and fewer than ten rehabilitation therapists per 100,000 population in China (*Committee NHaW, 2021*). Consequently, a substantial imbalance exists between the supply and demand for rehabilitation services in the country.

Knowledge graph, as a state-of-the-art technology, plays a pivotal role in transforming complex unstructured data into a structured "entity-relationship-entity" format (*Ji et al., 2021*). This transformation facilitates downstream tasks such as question and answer systems, decision systems, and information management systems. Leveraging the knowledge graph to organize rehabilitation data enables us to accelerate the advancement of the rehabilitation healthcare system, enhance the training of rehabilitation doctors, and enhance national awareness of rehabilitation, ultimately alleviating the shortage of rehabilitation resources in the country. As a crucial component in knowledge graph construction, named entity recognition (NER) plays a vital role in automatically extracting predefined entities from extensive and intricate texts, thereby facilitating the structuring of information (*Li et al., 2020a*). Earlier NER is primarily performed based on rules and dictionaries (*Fukuda et al., 1998*; *Keretna, Lim & Creighton, 2014*). While these methods demonstrate high accuracy, their heavy reliance on expert-defined rules hinders their extensibility. Later, the technology of machine learning break through and scholars start to apply statistical methods to NER, such as hidden Markov models, support vector machines (SVM), conditional random fields (CRF) (*Keretna et al., 2014*; *Lei et al., 2014*), *etc*. These methods demonstrate excellent performance with the support of rich corpora. Nevertheless, constructing such corpora requires substantial manual effort, and the *corpus*-specific features pose challenges in terms of portability and generalizability. With the enhancement of computer computing power and the surge of massive data, deep learning with powerful generalization ability has gradually become a research point in the field of named entity identification.

At present, apart from some voluntary forms of rehabilitation communities on the Internet such as, Hong Kong Society for Rehabilitation (https://www.rehabsociety.org.hk/zh-hans/) and Medical Pulse (https://www.medlive.cn/), no unified research in the field of rehabilitation medicine has been found for the time being. Scattered information about rehabilitation on web pages lacks credibility, so we chose to extract reliable rehabilitation information from thesis literature and books. We first search for CNKI literature (https://www.cnki.net) and authoritative guide books to build a *corpus* in the field of rehabilitation medicine in a comprehensive and multifaceted way, and use the annotation consistency principle to verify the accuracy of the *corpus*. In terms of model selection, the BERT

demonstrates robust linguistic representation and feature extraction capabilities, and Span can handle long medical entities and nested entities very well. Hence we combine the features of BERT and span to construct the BERT-Span model for the named entity recognition task in the rehabilitation domain. The experimental results indicate that the model achieves a higher F1 score in named entity identification studies in rehabilitation medicine compared to several baseline models.

The main contributions of this article are as follows:

(1) In view of the complexity and diversity of rehabilitation medical texts and the remoteness of vocabulary, we have formulated corresponding annotation specifications, adopted BIO annotation method, and have formed a certain scale of *corpus* in rehabilitation medicine.

(2) Combined with the results of annotation and the knowledge in the field of rehabilitation medicine, we propose the BERT-Span model that is more suitable for the task of Chinese named entity recognition in rehabilitation medicine, which fills the gap in NER in the field of rehabilitation medicine to a certain extent.

# RELATED WORK

## Basic neural network

Initially, convolutional neural networks (CNN), recurrent neural networks (RNN), Bi-directional long short-term memory (BiLSTM) and other basic neural networks can achieve good recognition results in NER tasks (*Li et al., 2021*; *Schuster & Paliwal, 1997*; *Zhang et al., 2015*). *Dong et al. (2016)* propose a multi-classification method for Chinese electronic medical records based on CNN, and achieve better results than traditional machine learning methods such as SVM and CRF in the self-constructed *corpus*. Later, *Dong et al. (2017)* propose to extract entities from Chinese electronic medical records using a multi-task bidirectional RNN model. It can further improve the performance of NER by determining the parameter weight through multi-layer bidirectional RNN. *Xu et al. (2018b)* train the BiLSTM-CRF model in the supervised *corpus* for medical named entity recognition. *Xu et al. (2018a)* also propose a similar BiLSTM-CRF model to specifically identify disease entities, which reaches 86.20% of F1 in NCBI disease data set. *Qin & Zeng (2018)*, on the basis of BiLSTM-CRF, use character based word embedding and continuous Bag-of-Words embedding (CBOW) as the input of the model to capture more representations. *Florez et al. (2018)* also improve the input of the BiLSTM-CRF model. They connect character level representation and part of speech features to form a comprehensive word representation, and then input the BiLSTM network to identify medical entities from clinical records.

## Neural network integrating attention mechanism

With the development of the research, the simple neural network model has exposed some problems. For example, global features can not be obtained, and information between long sentences is seriously missing. The concept of weight distribution in attention mechanism can effectively solve such problems. Some scholars commence adding attention mechanism to BiLSTM-CRF for named entity recognition. *Ji et al. (2019)* add the attention

mechanism to the BiLSTM-CRF model to conduct medical NER research on Chinese electronic medical records, and obtain 90.82% of F1 scores in the open dataset. *Wu et al. (2020)* use attention mechanism to complete NER and intention analysis tasks in medical problems based on the BiLSTM-CRF model. *Li et al. (2019a)* combine attention mechanism with neural network (BiLSTM ATT-CRF), and introduce medical dictionaries and part of speech features, which confirmed the effectiveness of attention mechanism in obtaining long sentence information and text features. *Wu et al. (2019)* propose the Att-BiLSTM-CRF model to capture long-range dependencies using a self-attentive mechanism that combines a new character-level representation and POS annotation information to capture the semantic information of the input sentences, taking into account the own characteristics of Chinese. In order to better explain the meaning of Chinese characters, *Yin et al. (2019)* use CNN to extract radical-level features on the basis of the BiLSTM-CRF model, aiming to capture the internal relevance of characters, and use the self-attention mechanism to capture the dependency between characters. *An et al. (2022)* add the multi-head self-attention mechanism on the basis of BiLSTM-CRF and proposed the MUSA BiLSTM-CRF model, which further improve the NER in terms of global information and multi-level semantic information acquisition. In addition, some scholars try to improve the BiLSTM layer of traditional models to improve the model recognition effect. *Zhao et al. (2019)* propose a convolutional neural network (AT lattice LSTM-CRF) based on attention mechanism. Lattice-LSTM integrates character level information and word-level information, making up for the defect of single information. *Li et al. (2020b)* put forward the ELMo lattice LSTM-CRF model, which uses lattice LSTM to obtain word and character information. The variant ELMo model uses Chinese characters as input to learn context information in specific fields. Some scholars use Transformer to code instead of BiLSTM. *Wan et al. (2020)* propose a method based on ELMo-ET-CRF model. Compared with ELMo-LSTM-CRF model, ELMo-ET-CRF has better performance.

## Transfer learning

However, numerous tasks in NER require a large amount of annotated data, and NER in some new fields do not meet these conditions. Transfer learning focuses on storing existing problem solving models and using them on other different but related problems (*Pan & Yang, 2010*). Therefore, some scholars put forward the idea of transfer learning to complete this task. *Bhatia, Arumae & Busra Celikkaya (2020)* learn the appropriate parameter sharing scheme between the source dataset and the target dataset through the dynamic transmission network (DTN), so as to achieve the best NER result on a smaller target dataset. *Dong et al. (2019)* propose to train the shallow bidirectional RNN network on the general domain data set first, then migrate the weights of the trained parameters to the deep bidirectional RNN network, and train again on the Chinese electronic medical record. *Gong, Zhang & Chen (2020)* use a *corpus* of the same domain to pre-train word embedding and subsequently fine-tune the pre-training model of the target *corpus* using the model parameters obtain from the training to overcome the challenge of a small annotated *corpus* of Chinese clinical entities. Later, with its strong language representation ability and feature extraction ability, BERT make outstanding achievements in the field of named

entity recognition (*Devlin et al., 2018*). *Xue et al. (2019)* use BERT to extract the word vector of the text as the input of the subsequent model. *Zhang et al. (2019)* propose the BERT-BiLSTM-CRF pre-training language model. The experimental results show that compare with the previous model, the BiLSTM-CRF model combines with BERT has the most outstanding effect in Chinese electronic medical record entity recognition. *Li, Zhang & Zhou (2020)* use unlabeled Chinese clinical text to pre-training BERT model to learn domain specific knowledge, and use BiLSTM and CRF to extract text features and decode predictive tags respectively. *Liu et al. (2022)* propose the Med BERT-Span FLAT model, in which Med BERT optimizes the representation of long medical entities, and Span-FLAT replaces the traditional CRF and has a better effect on nested entity annotation.

In summary, the algorithmic technology employed for NER tasks is steadily advancing, and its application in the medical field has yielded high accuracy results. However, in certain specialized fields, the presence of intricate and less familiar vocabulary, coupled with limited available corpora, poses challenges to named entity recognition. Currently, no relevant NER studies have been found in the field of rehabilitation medicine. With the maturity of migration learning, the recognition of named entities in these new fields can no longer require a large amount of annotated *corpus* to be prepared in advance to have a good recognition effect. Therefore, in this article, we focus on the task of named entity recognition in the field of Chinese rehabilitation medicine. In view of the fact that there is no *corpus* in this field at present, we collect rehabilitation data from CNKI literature database, rehabilitation books *etc.* to build a small-scale rehabilitation *corpus*, and verify the reliability of the *corpus* by the annotation consistency principle. Aiming at the characteristics of entities in the field of rehabilitation medicine, we construct the BERT-Span model to solve the recognition problem of long and nested entities, and verify the validity of the model by comparing with the classical baseline model.

## METHODS

### Data preparation

Portions of this text were previously published as part of a preprint (*Zhong et al., 2023*). Stroke is one of the most important public health issues of global concern and is also a hot topic in the field of rehabilitation medicine (*Zhou et al., 2019*). Our research will also focus on stroke in rehabilitation. Experimental data for this article are mainly from stroke rehabilitation guidelines and medical literature. Considering the richness and authority of the data, we select CNKI (https://www.cnki.net/) as our retrieval database. Set keywords "stroke", "rehabilitation" and corresponding synonyms in CNKI advanced search, and select "CSSCI", "core journals" and "CSCD" literature from 2011–2020. After manual screening, some irrelevant and repetitive literatures were removed, and 160 literatures were derived as partial data sources. Finally, a total of 10,000 corpora are collected from rehabilitation guides and CNKI literature, totaling 612,807 words.

Since there is no unified standard for naming entity identification in the field of rehabilitation medicine, on the basis of our previous experience in entity labeling in the field of medicine and the characteristics of texts in the field of rehabilitation, we divide rehabilitation entities into: "dysfunction and performance", "rehabilitation assessment",

{"text": "Early identification of patients at risk for pressure ulcers and collaboration between caregivers to prevent pressure sores is essential. Patients at risk for pressure ulcers may have the following conditions: (1) impaired voluntary mobility; (2) diabetes mellitus; (3) peripheral vascular disease; (4) urinary incontinence; (5) high or low body mass index; (6) sensory impairment; and (7) concurrent other malignant diseases." , "label": {"Dysfunction and performance": {"Pressure sores": [[4, 5], [23, 24], [30, 31]], "Impaired voluntary mobility": [[46, 53]], "diabetes mellitus": [[56, 58]], "peripheral vascular disease": [[61, 66]], "urinary incontinence": [[69, 72]], "high or low body mass index ": [[75, 83]], "sensory impairment": [[86, 89]]}}}

**Figure 1 A *corpus* diagram in a JSON file.**

"rehabilitation methods", "rehabilitation equipment", "body" and "drugs" six categories. Among them, the first four categories are summarized from a large number of rehabilitation literature, and the last two categories are obtained by combining the common points of rehabilitation medicine and clinical medicine.

## *Corpus* construction

Referring to the existing medical *corpus*, we have built a *corpus* in the field of rehabilitation medicine. In the context of *corpus* annotation, the article follows the following specifications for the annotation of four types of entities in rehabilitation medicine: non-overlapping annotation, full foreign language vocabulary not marked, and special symbols and conjunctions can be annotated.

In order to ensure the accuracy of annotation, we primarily relies on manual annotation and verification when creating the rehabilitation *corpus*. We conduct three rounds of manual annotation and verification, and extract 500, 1,250, and 1,250 corpora respectively, until there are no entities that are difficult to regulate boundaries or ambiguous categories in the annotation process. For the 2,500 corpora in the last two rounds, we randomly select 500 corpora and submit them to the second person to label independently. The results of the two labels on the 500 corpora are compared and evaluated according to the Inter annotator Agreement (IAA) principle (*Roberts et al., 2009*). After calculation, the F1 value of these 500 rehabilitation corpora reaches 85%, so it is believed that the rehabilitation *corpus* independently constructed in this article is reliable.

Based on the above standard principles and workflow of annotation, we complete annotation work for 2,500 corpora. Find out the entities that appear in each sentence, and mark the category and location of each entity; Correct the wrong annotation form and overlapping annotation, and realize the error detection and omission of manually annotated entities; The manual annotation results are then evaluated. The format of a certain *corpus* in the final JSON file is shown in Fig. 1:

To sum up, we collects 10,000 corpora in the field of rehabilitation medicine, and marks 2,500 of them manually. The number of entities in six categories in the 2,500 corpora is shown in Table 1:

| Entity type | Dysfunction and performance | Rehabilitation assessment | Rehabilitation methods | Rehabilitation equipment | Body | Drugs |
|---|---|---|---|---|---|---|
| Number of entity | 3,881 | 2,751 | 3,069 | 445 | 2,312 | 130 |

Table 1 Number of entities in *corpus*.

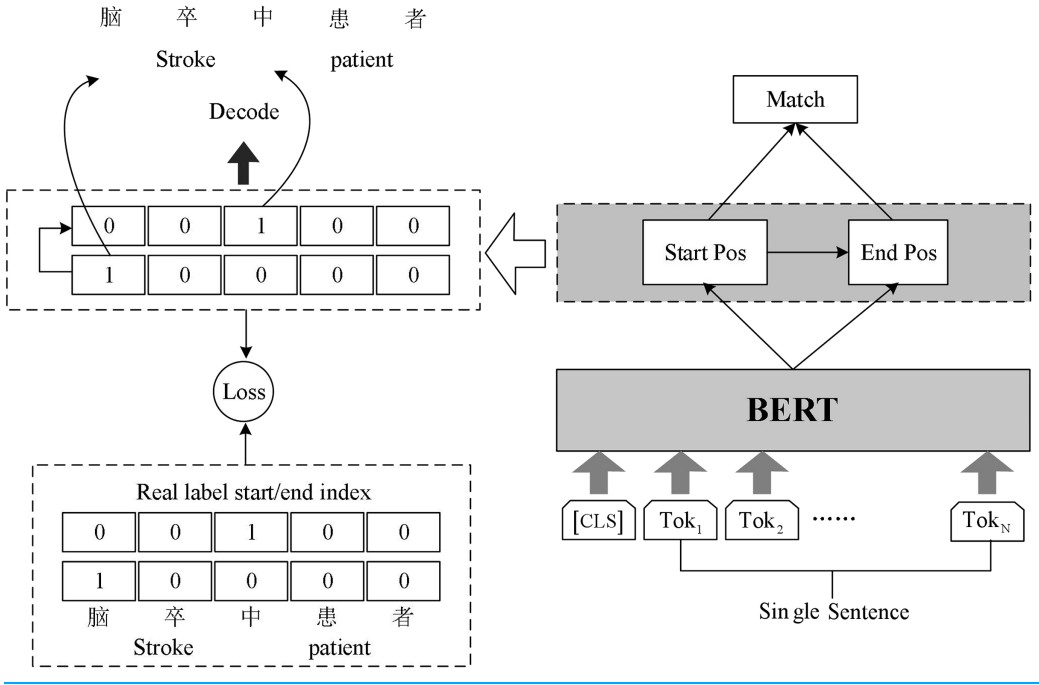

**Figure 2** Schematic diagram of BERT-Span model.

## BERT-Span model

Chinese rehabilitation medicine field vocabulary is rare, word structure is complex, long noun phrases and nested noun phrases are often found in the context. In order to identify these entity phrases more accurately and improve the accuracy of named entity recognition, we build BERT-Span model for named entity recognition task in rehabilitation medicine. The model structure is shown in Fig. 2.

The sentences in the text are vectorized and fed into BERT, which leverages the masked language model (MLM) pre-training technique. By employing multiple deep bi-directional Transformers, BERT generates deep bi-directional language representation vectors that capture contextual information. These vectors serve as text features and enable BERT to extract meaningful representations from the input sentences.

There are a greater number of long medical entities and nested medical entities in rehabilitation medicine texts. Sequence annotation can only be assigned to the BIO tag corresponding to each token, and becomes insufficient when appearing nested entities. Some researchers propose an idea referring to machine reading comprehension (MRC) model framework technology (*Li et al., 2019b*), specifically dealing with nested entity recognition. We follow this inspiration and use Span model to implement the annotation task.

The Span model uses double sequence tagging in the decoding process. One sequence is used to mark the start position of all entities, and the other sequence is used to mark the end position of all entities. Two linear layers are used for decoding operations. Decoding operation using two linear layers to extract the head and tail part of the entity.

Since there may be multiple entities belonging to the same category in the input text, the model will get several start indexes and end indexes simultaneously during decoding. Hence the model needs to match the decoded start index and end index appropriately. We complete the matching of the start and end index of the entities using the binary classifier. The classifier inputs the possible start and end index of the entity in the sentence and outputs 0 or 1 to indicate whether it matches or not. The formula in Span is as follows:

$$P_{start} = soft \max eachrow(E \cdot T_{start}) \in R^{n \times 2}$$
$$P_{end} = soft \max eachrow(E + one - hot(P_{start}) \cdot T_{end}) \in R^{n \times 2}$$
$$\hat{I}_{start} = \left\{ i \mid \arg \max(P_{start}^{(i)}) = 1, i = 1, 2 \cdots, n \right\}$$
$$\hat{I}_{end} = \left\{ j \mid \arg \max(P_{end}^{(j)}) = 1, j = 1, 2 \cdots, n \right\}$$
$$P_{i_{start}, j_{end}} = sigmoid\left(m \cdot concat\left(E_{i_{start}}, E_{j_{end}}\right)\right) \tag{1}$$

Obtain the representation matrix E from the output of BERT, the model first predicts the probability of each token as the starting index $p_{start}$, where $T_{start}$ is the weight of learning. Each row of $p_{start}$ represents the probability distribution of each index as the starting position of the entity for a given query. The $p_{start}$ unique hot code introduced in the $P_{end}$ calculation process adds the starting position information to the ending sequence. Apply argmax function to each row of and to get the prediction index that may be the start or end position, $\hat{I}_{start}$ and $\hat{I}_{end}$, where the $P^{(i)}$ superscript i represents the ith row of the matrix. Given random starting index $i_{start} \in \hat{I}_{start}$ and ending index $j_{end} \in \hat{I}_{end}$, the matching probability of the two indexes is predicted by sigmoid function binary classification model, where m is the weight obtained from learning.

The loss function L consists of three parts: the entity start position loss function $L_{start}$, the entity end position loss function $L_{end}$, and the entity interval loss function $L_{span}$. $L_{start}$ denotes the deviation of the predicted entity start position from the true entity start position. $L_{end}$ denotes the deviation of the predicted entity end position from the true entity start position. $L_{span}$ denotes the deviation of the predicted entity location interval from the true entity location interval. The loss function for each part uses a cross-entropy loss function, where *P* denotes the model prediction probability and Y is the sample label. The weighted sum of these three loss components is the final overall loss value, where α, β, and γ are the model hyperparameters. The specific formula is as follows:

$$L_{start} = CE(P_{start}, Y_{start})$$
$$L_{end} = CE(P_{end}, Y_{end})$$
$$L_{span} = CE\left(P_{start, end}, Y_{start, end}\right)$$
$$L = \alpha L_{start} + \beta L_{end} + \gamma L_{span} \tag{2}$$

## RESULTS AND DISCUSSION

In this section, we compare our propose model with several baseline models and analyze the superiority of our approach from different perspectives. For our model, we try to explore the properties of entities in rehabilitation medicine texts from the perspective to further explore the performance of the model for text processing in the Chinese rehabilitation medicine domain. BERT-softmax and BERT-BiLSTM-CRF are the most widely used baseline models in the field of named entity recognition at the present time, and are also the comparison models chosen for this experiment. During the experiment, we use Adam optimizer with initial learning rate of $2 \times 10^{-5}$ and warmup_proportion of 0.15. Train batch size is 8 and the train epoch is 20, and the ratio of training set and test set is set to 9:1. The rest of the hyperparameters are default values. Our code and data are released at https://github.com/jamesXuan/BERT-Span.

We use the most commonly used evaluation metrics in the field of named entity recognition: precision rate $P$, recall rate R and F1 value, where, precision rate $P$ refers to the probability of the actual correct samples among all the samples predicted to be correct; recall rate R refers to the probability of the predicted samples being correct among the actual correct samples; F1 value is the summed average of recall rate and precision rate, and the calculation formula is given in:

$$P = \frac{TP}{TP + FP}$$
$$R = \frac{TP}{TP + FN}$$
$$F1 = \frac{2PR}{P + R} \tag{3}$$

### A comparative experiment of different models in the same *corpus*

Figure 3 illustrates the performance of the BERT-Span compared to the BERT-softmax, BERT-BiLSTM-CRF models on a self-built rehabilitation *corpus* for the rehabilitation named entity recognition task. To mitigate the potential impact of a small dataset leading to significant experimental errors, a 5-fold cross-validation is employed.

It can be seen that with the auxiliary of the pre-trained language model BERT, these models generally have better results for the recognition of named entities in rehabilitation medicine. On this basis, BiLSTM-CRF is advantageous for entity recognition compared to softmax, where the former focuses more on the global information of the sequence. BERT-BiLSTM-CRF outperforms BERT-softmax in terms of recall (R) and F1-score metrics. In addition, there are long medical entities and nested medical entities in the field of rehabilitation medicine, and CRF is difficult to distinguish them because of the limitation of its own structure and cannot play a good effect, while Span can solve this problem well, so the BERT-Span model achieves the optimal effect for the recognition of named entities in the field of rehabilitation medicine.

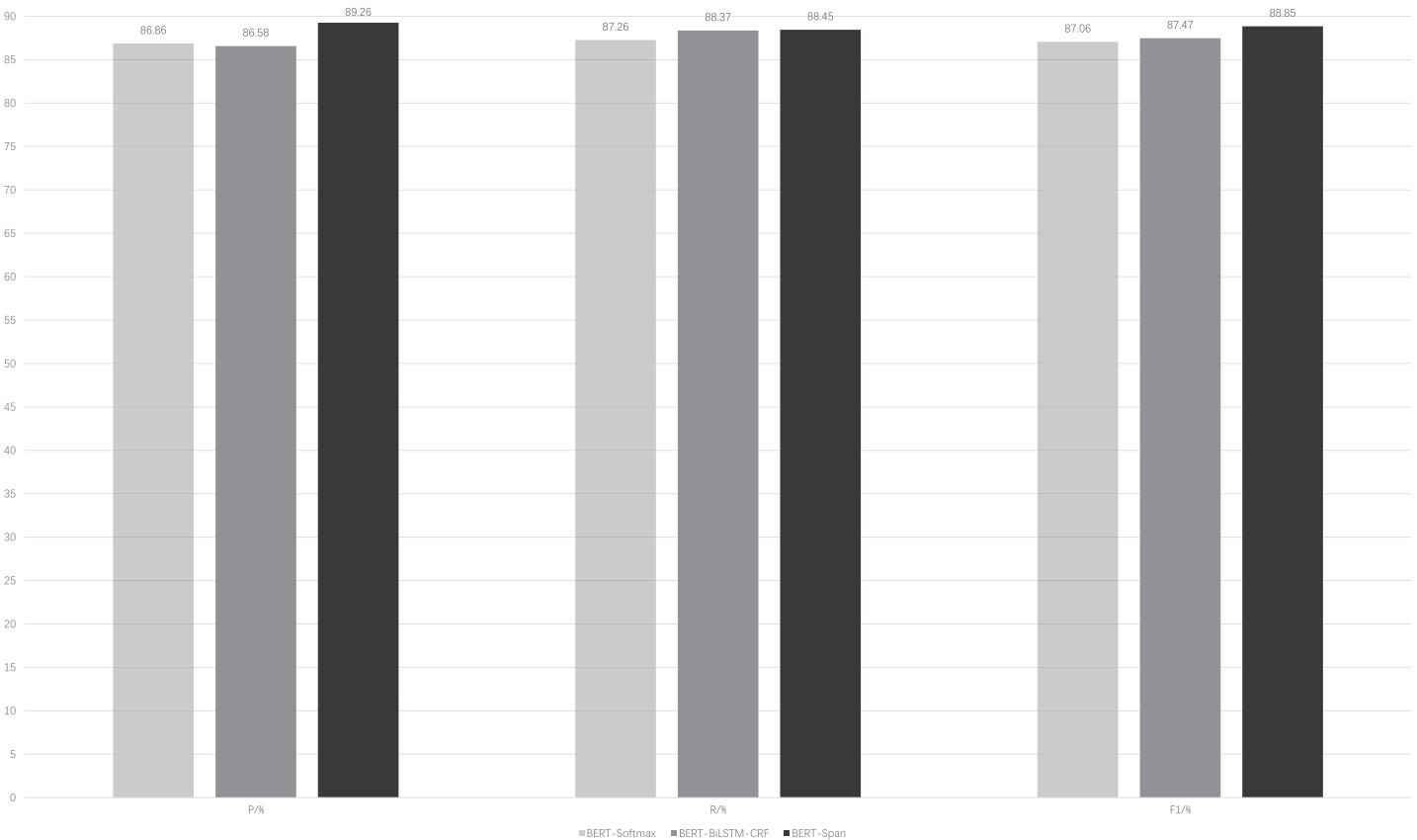

**Figure 3 Comparison of model performance for the same data set.** BERT-softmax, bidirectional encoder representation from transformers-softmax; BERT-BiLSTM-CRF, bidirectional encoder representation from transformers-bi-directional long short-term memory-conditional random fields; BERT-Span, bidirectional encoder representation from transformers-span.

## A comparative experiment on different corpora

Table 2 illustrates the effect of different *corpus* sizes on model performance. The BERT-Span model was first used to predict the annotation of 7,500 unannotated data, and a total of 2,500 manually annotated data and 7,500 model annotated data were obtained. Among the 2,500 manually labeled data, 1,250 data were randomly selected as a common test set for two sets of experiments. The first set of experiments uses the remaining 1,250 manually labeled data as the training set; the training set of the second set of experiments consists of the first training set and 7,500 model labeled data.

From the table, it is clear that the performance of all three models is significantly improved after increasing the data size of the training set. In terms of F1 metrics, BERT-softmax increases by 1.7%, BERT-BiLSTM-CRF increases by 1.37%, and BERT-Span has the highest increase of 1.8%. The amount of data is still an important factor limiting the performance of the model. Notably, the BERT-Span model consistently achieves optimal performance in the named entity identification task within rehabilitation medicine, both before and after the dataset expansion.

**Table 2 Effect of training data size on model performance.**

| Groups | Model | P | R | F1 |
|---|---|---|---|---|
| 1 | BERT-Softmax | 82.40 | 85.12 | 83.74 |
| | BERT-BiLSTM-CRF | 82.68 | 85.92 | 84.27 |
| | BERT-Span | 85.58 | 83.92 | 84.75 |
| 2 | BERT-Softmax | 84.16 | 86.77 | 85.44 |
| | BERT-BiLSTM-CRF | 84.42 | 86.89 | 85.64 |
| | BERT-Span | 86.01 | 87.11 | 86.55 |

**Table 3 Entity recognition performance statistics.**

| Entity | P | R | F1 |
|---|---|---|---|
| Dysfunction and performance | 93.19 | 92.95 | 93.07 |
| Rehabilitation assessment | 88.80 | 85.38 | 87.06 |
| Rehabilitation methods | 87.98 | 87.98 | 87.98 |
| Rehabilitation equipment | 85.29 | 85.29 | 85.29 |
| Body | 86.78 | 87.87 | 87.32 |
| Drugs | 100 | 83.33 | 90.91 |

## Exploration of the recognition accuracy of different entities

Table 3 presents a comprehensive analysis of the BERT-Span model's performance in identifying the six categories of entities within rehabilitation medicine.

As can be seen from the table, there are considerable differences in the recognition effects of different entities. The identification accuracy of rehabilitation assessment, rehabilitation methods, rehabilitation equipment, and body was low, with F1 values below 90%, where the identification effect of rehabilitation equipment was the worst, with an F1 value of 85.29%. The recognition accuracy of dysfunction and performance, and drug entities was higher, with F1 values over 90%, and "dysfunction and performance" had the best recognition effect, with an F1 value of 93.07%.

The size of the dataset and the distribution of the labels have an impact on the final prediction results of the model. Therefore, we try to explain the variability of identification results between different classes of entities in terms of these two aspects.

We collated the number of each entity in the dataset, the number of low-frequency entities (the number of entity occurrences is 1), and the percentage of low-frequency entities in the total number of entities, as shown in Table 4.

We can see that "dysfunction and performance" has the lowest percentage of low frequencies. The model can extract the feature information of these entities more accurately and give correct prediction results. The highest percentage of low frequencies for "rehabilitation equipment" indicates that most of the data of "rehabilitation equipment" entities only appear once in the data set, and the model cannot obtain their feature information well enough to give high prediction accuracy. Meanwhile, the BERT

**Table 4 Entity number and frequency statistics.**

| Entity type | No. of low-frequency entities | No. of entities | Proportion |
|---|---|---|---|
| Dysfunction and performance | 666 | 3,881 | 17% |
| Rehabilitation assessment | 551 | 2,751 | 20% |
| Rehabilitation methods | 866 | 3,069 | 28% |
| Rehabilitation equipment | 144 | 445 | 32% |
| Body | 472 | 2,312 | 20% |
| Drugs | 42 | 130 | 32% |

pre-training model we use is based on the general clinical medicine *corpus*, while the two types of entities "drug" and "body" exist in clinical medicine, and they can obtain some feature information in advance. The four entities of "dysfunction and performance", "rehabilitation assessment", "rehabilitation methods", and "rehabilitation equipment" exist only in the field of rehabilitation medicine and can only learn feature information during model training. Therefore, although the low frequency of "drug" is as high as 32% as the low frequency of "rehabilitation equipment", the drug has learned some of its characteristic information in advance, so it still has a high recognition effect. It can be expected that dealing with the low-frequency word problem will have further improvement on the performance of the model.

## CONCLUSIONS

Named entity recognition in rehabilitation medicine is an important step in many rehabilitation text information extraction tasks. In order to advance the research of named entity recognition in rehabilitation field, we collect data from various ways, discuss the scheme with rehabilitation physicians, and build a rehabilitation *corpus*, which solves the problem that there is no ready-made *corpus* available in the field of rehabilitation medicine. To avoid the influence of long medical entities and nested entities on Chinese named entity recognition in rehabilitation medicine, we proposed the Bert-Span model, using span tokens instead of traditional CRF sequence tokens. The experimental results demonstrate that the BERT-Span model is more suitable for named entity recognition studies in rehabilitation medicine compared with the BERT-Softmax and BERT-BiLSTM-CRF baseline models.

However, there are some limitations of our study. On the one hand, we have only validated the performance of the Chinese rehabilitation medicine named entity recognition task on a self-built *corpus*, and its effectiveness on Chinese NERs from other rehabilitation datasets needs further validation. On the other hand, the performance of the model on named entity recognition tasks in other language rehabilitation domains is unclear.

In the future, we would apply our model to the actual rehabilitation medicine text annotating tasks, so as to help build the knowledge graph of rehabilitation medicine. Moreover, we plan to extend our approach to other relevant rehabilitation medicine text

datasets to further validate and improve the model's scalability and generalization capabilities.

## ACKNOWLEDGEMENTS

The authors would like to thank all members of the research team for their technical support during the research activities.

### Funding

The authors received no funding for this work.

### Competing Interests

The authors declare that they have no competing interests.

### Author Contributions

- Jinhong Zhong conceived and designed the experiments, performed the experiments, analyzed the data, performed the computation work, authored or reviewed drafts of the article, and approved the final draft.
- Zhanxiang Xuan conceived and designed the experiments, performed the experiments, performed the computation work, authored or reviewed drafts of the article, and approved the final draft.
- Kang Wang analyzed the data, prepared figures and/or tables, and approved the final draft.
- Zhou Cheng conceived and designed the experiments, prepared figures and/or tables, and approved the final draft.

### Data Availability

Code and raw data are available at GitHub and Zenodo:

https://github.com/jamesXuan/BERT-Span.

Zhanxiang Xuan. (2023). BERT-Span dataset [Data set]. Zenodo. https://doi.org/10.5281/zenodo.8088554.

The experimental data in this article are from the rehabilitation guide of stroke and medical literature of CNKI platform (https://www.cnki.net/).

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
