# Peer review of "A BERT-Span model for Chinese named entity recognition in rehabilitation medicine"

_PeerJ Computer Science, doi:10.7717/peerj-cs.1535_

## Round 0.1 · original submission · Major Revisions

The reviewers have substantial concerns about this manuscript. The authors should provide point-to-point responses to address all the concerns and provide a revised manuscript with the revised parts being marked in different color.

Reviewer 1 ·

Basic reporting

It is evident that there is room for improvement in terms of the English language usage employed throughout the manuscript. To elaborate, certain sentences within the method section exhibit incorrect verb tenses, necessitating a revision to ensure grammatical accuracy. Furthermore, a comprehensive proofreading to address other instances of imprecise language would contribute to the overall clarity and readability of the manuscript.

It is recommended for the authors to acknowledge and address the limitations of their work within the discussion section, as this will provide a comprehensive understanding of the study's scope and potential areas for future research.

Experimental design

Within the data preparation section, the authors explicitly state the selection of "stroke" and "rehabilitation" as keywords to retrieve relevant literature from CNKI. However, it is essential to expand upon this explanation by including additional synonyms for "stroke" and "rehabilitation" that were considered during the keyword selection process. Furthermore, clarifying the rationale behind exclusively choosing these two keywords and utilizing CNKI as the sole data source, without considering alternatives like WANFANG, would strengthen the research methodology and provide readers with a more comprehensive understanding.

Validity of the findings

To promote the practicality and reproducibility of the research, it would be highly advantageous if the authors could make the code and annotation corpora openly accessible. By providing unrestricted access to these resources, researchers and practitioners in the field will be able to utilize and build upon the results presented in the paper, thus fostering collaboration and advancing the state-of-the-art knowledge in the domain.

The discussion section requires attention in terms of its content and organization. It is essential to thoroughly discuss and interpret the results, highlighting the unique aspects of the method in comparison to another existing research. This will provide valuable insights and strengthen the manuscript's contribution to the field.

Cite this review as

Reviewer 2 ·

Basic reporting

1.The manuscript would greatly benefit from a thorough reevaluation to address small errors. For example, it is advisable to consistently utilize the full names when introducing abbreviations, such as "Bidirectional Encoder Representations from Transformers (BERT)," instead of using the abbreviation alone. Additionally, it is crucial to rectify the issue of missing citations, such as BERT and CNN, which are absent from the introduction section. These errors persist throughout various sections of the manuscript and should be carefully revised.
2.A significant concern arises from the content presented in the manuscript from line 93 to 174, which predominantly consists of a literature review. To align with standard research practices, it is crucial to separate this content from the introduction section and designate it as a distinct and appropriately titled section, such as "Related Work." This reorganization will enhance the logical structure of the manuscript and assist readers in navigating the research more effectively.

Experimental design

1.The figure would greatly benefit from improvement to enhance reader comprehension. In Figure 2, it is recommended to include English labels or provide translations for better understanding. Additionally, in Figure 3, it is crucial to incorporate the full names of each abbreviation to avoid ambiguity and ensure clarity for readers.

Validity of the findings

1. In the discussion section, Table 3 should be relocated to the results section instead of being placed within the discussion section, as it directly presents empirical findings.

Additional comments

1.An area that requires immediate attention is the abstract, which currently lacks a critical component. Specifically, the abstract encompasses three distinct parts but notably omits a discussion or conclusion segment. To rectify this issue, it is strongly recommended that the author includes a succinct yet informative discussion or conclusion section within the abstract. This addition will enable readers to glean the main findings and implications of the research directly from the abstract, further enhancing the manuscript's overall accessibility and impact.

Cite this review as

Reviewer 3 ·

Basic reporting

The paper presents a compelling argument for the necessity of rehabilitation medical facilities in China. To address this demand, the authors propose a novel approach that combines the powerful BERT model with Span, leveraging their strengths in linguistic representation, feature extraction, and handling long and nested medical entities. This innovative combination holds great potential for enhancing the effectiveness and efficiency of rehabilitation medical systems, ultimately meeting the growing demand in this domain.

1.To avoid confusion among readers, it would be beneficial to explicitly mention at the beginning of the paper that it focuses on a language model specifically designed for Chinese. This clarification will help readers understand why the discussion revolves around rehabilitation in China.

2. Furthermore, it would be advantageous for the author to provide additional details about their work before delving into the methods section.

Experimental design

How are the model superparameters selected?

Validity of the findings

Figure 3 only showed the performance for one same data set but could not meet with the statistical significance, would it be better to try multiple datasets, maybe using cross-validation?

Cite this review as

---

## Round 0.2 · accepted · Accept

All the concerns have been addressed. I would suggest accepting this manuscript.

Reviewer 1 ·

Basic reporting

No further concern

Experimental design

No further concern

Validity of the findings

No further concern

Cite this review as

Reviewer 3 ·

Basic reporting

The authors changed the paper titles and also clarified unclear statements, such as a summary of their work before the method parts in the revision version. It should meet the requirement of basic reporting.

Experimental design

The authors shared the code and explained the details of selecting the hyperparameters.

Validity of the findings

The authors tested on 5-fold cross validation on a single dataset, and the results may be acceptable.

Cite this review as